# Entropy-Based Solutions for Ecological Inference Problems: A Composite Estimator

**DOI:** 10.3390/e22070781

**Published:** 2020-07-17

**Authors:** Rosa Bernardini Papalia, Esteban Fernandez Vazquez

**Affiliations:** 1Department of Statistical Sciences, University of Bologna, 40126 Bologna, Italy; rossella.bernardini@unibo.it; 2REGIOlab and Department of Applied Economics, University of Oviedo, 33003 Oviedo, Spain

**Keywords:** ecological inference, generalized cross entropy, distributional weighted regression, matrix adjustment

## Abstract

Information-based estimation techniques are becoming more popular in the field of Ecological Inference. Within this branch of estimation techniques, two alternative approaches can be pointed out. The first one is the Generalized Maximum Entropy (GME) approach based on a matrix adjustment problem where the only observable information is given by the margins of the target matrix. An alternative approach is based on a distributionally weighted regression (DWR) equation. These two approaches have been studied so far as completely different streams, even when there are clear connections between them. In this paper we present these connections explicitly. More specifically, we show that under certain conditions the generalized cross-entropy (GCE) solution for a matrix adjustment problem and the GME estimator of a DWR equation differ only in terms of the a priori information considered. Then, we move a step forward and propose a composite estimator that combines the two priors considered in both approaches. Finally, we present a numerical experiment and an empirical application based on Spanish data for the 2010 year.

## 1. Introduction

Ecological inference (EI) is the process of drawing conclusions about individual-level behavior from aggregate (historically called “ecological”) data, when no individual data are available. Situations where the only available data are aggregated at a level other than the level of interest are quite common in many application fields. This is the typical setting for Ecological Inference [1,2,3], Cross-level Inference [4,5], Small Area Estimation [6], or disaggregation methods [7]. The basic idea is that, in order to study the behavior of the individuals (or sub-groups of individuals), a microeconomic analysis ought to be carried out using fairly localized individual data, and data which are aggregated by areal units may be used in order to investigate the behavior of the individuals comprising those units. In this paper, we specifically refer to the process of drawing conclusions about individual-level behavior from aggregate data, when no individual data are available or when individual data are incomplete. In this inferential context, one problem is that many different possible relationships at the individual (or subgroup) level can generate the same observations at the aggregate (or group) level [8]. In the absence of individual (or subgroup) level measurements (in the form of survey data), such information needs to be inferred. Estimates of the disaggregated values for the variable of interest can be inferred from aggregate data by using appropriate statistical techniques. However, in many situations, given that the micro-data of interest are not available, the accuracy of any predicted value cannot be verified. This research focuses on the estimation on disaggregated indicators by subclasses. Assume that we have an indicator, $y_{i\cdot }$, that is observable across the different areas i=1,…,T. Our objective is to disaggregate it into an indicator $y_{ij}$ for the j=1,…,K different sub-categories (or sub-areas) that conform each class (or area) *i*. The information available for this inference exercise, together with the indicator $y_{i\cdot }$, is another disaggregated indicator $x_{ij}$ that is related to the target indicator $y_{ij}$. This paper approaches this estimation problem in an attempt to unify two estimation strategies and it is organized as follows. Section 2 explains the main features of the matrix-adjustment following the ideas of the Generalized Cross Entropy (GCE) estimation introduced in [9], whereas in Section 3 the basis of the Distributionally Weighted Regression (DWR) estimation are explained. Section 4 studies these two strategies under a common approach and propose a composite prior estimator in line with the Data Weighted Prior (DWP) proposed in [10,11]. The comparative performance of the three techniques is evaluated by means of a numerical experiment in Section 5. Finally, Section 6 presents the main conclusions of the paper.

## 2. Matrix-Adjustment and Distributionally Weighted Regression Problems

Within the family of IT estimators, [10] proposed a general solution for the estimation problem described in the introduction basing on the minimization of the divergence between the target variable and some prior information. Following this approach, each indicator $y_{ij}$ is assumed as a discrete random variable that can take M different values. Defining a supporting vector (for the sake of simplicity assumed as common for all the $y_{ij}$) $z^{\prime }=\left[z_{1},z_{2},\ldots ,\;z_{M}\right]$ that contains the M possible realizations of the targets with unknown probabilities $p_{ij}^{\prime }=\left[p_{ij1},p_{ij2},\ldots ,\;p_{ijM}\right]$, $y_{ij}$ can be written as:(1)$$y_{ij}=\overset{M}{\underset{m=1}{\sum }}p_{ijm}z_{m}$$

Alternatively, this idea can be generalized in order to include an error term and define each $y_{ij}$ as:(2)$$y_{ij}=\overset{M}{\underset{m=1}{\sum }}p_{ijm}z_{m}+\varepsilon_{ij}$$

In such a case, we assume that the $y_{ij}$ elements are given from two sources: a signal that keeps the resemblance with the priors $x_{ij}$, plus a noise term ($\varepsilon_{ij}$). The noise components can be included in order to account for potential spatial heterogeneity and our uncertainty about the target variable. Basically, we represent uncertainty about the realizations of the errors treating each element $\varepsilon_{ij}$ as a discrete random variable with L≥2 possible outcomes contained in a convex set $v^{\prime }=\left\{v_{1},\ldots ,v_{L}\right\}$, which for the sake of simplicity will be assumed as common for all the $\varepsilon_{ij}$. We also assume that these possible realizations are symmetric around zero ($-v_{1}=v_{L}$). The traditional way of fixing the upper and lower limits of this set is to apply the three-sigma rule [12]. Under these conditions, each $\varepsilon_{ij}$ can be defined as:(3)$$\varepsilon_{ij}=\overset{L}{\underset{l=1}{\sum }}w_{ijl}v_{l};\;\forall i=1,\ldots ,T;\;j=1,\ldots ,K$$
where $w_{ijl}$ is the unknown probability of the outcome $v_{l}$ for the cell *ij*. Now, the $y_{ij}$ elements can be written as:(4)$$y_{ij}=\overset{M}{\underset{m=1}{\sum }}p_{ijm}z_{m}+\overset{L}{\underset{l=1}{\sum }}w_{ijl}v_{l}$$

The solution to the estimation problem is given by the minimization of the Kullback-Leibler divergence between the posteriors distributions $p^{\prime }$s and the a priori probabilities $q_{ij}^{\prime }=\left[q_{ij1},q_{ij2},\ldots ,\;q_{ijM}\right]$. The $q^{\prime }$s reflect the information we have on the indicators $x_{ij}$, which are somehow related to our target $y_{ij}$, being defined by the expression:(5)$$x_{ij}=\overset{M}{\underset{m=1}{\sum }}q_{ijm}z_{m}$$

The solution to the estimation problems is given by minimizing the KL divergence between the $p^{\prime }$s and the $q^{\prime }$s. If we do not have an informative prior, the a priori distributions are specified as uniform $\left(q_{ij}^{}=\frac{1}{M};\;\forall m=1,\ldots ,M\right)$, which leads to the GME solution. The uniform distribution is usually set as the natural prior $W^{0}$ for the error terms. Specifically, the constrained minimization problem can be written as: (6)$$\underset{p,W}{Min}D\left(p,W\|q,W^{0}\right)=\overset{M}{\underset{m=1}{\sum }}\overset{T}{\underset{i=1}{\sum }}\overset{K}{\underset{j=1}{\sum }}p_{ijm}ln\left(\frac{p_{ijm}}{q_{ijm}}\right)++\overset{L}{\underset{l=1}{\sum }}\overset{T}{\underset{i=1}{\sum }}\overset{K}{\underset{j=1}{\sum }}w_{ijl}ln\left(\frac{w_{ijl}}{w_{ijl}^{0}}\right)$$
subject to:(7)$$y_{i\cdot }=\overset{K}{\underset{j=1}{\sum }}\left(\overset{M}{\underset{m=1}{\sum }}p_{ijm}z_{m}+\overset{L}{\underset{l=1}{\sum }}w_{ijl}v_{l}\right)C_{\cdot j};\;i=1,\ldots ,T$$
(8)$$\overset{M}{\underset{m=1}{\sum }}p_{ijm}=\overset{L}{\underset{l=1}{\sum }}w_{ijl}=1;1\;\forall i=1,\ldots ,T;\;j=1,\ldots ,K$$

Restrictions (8) are just normalization constrains, whereas Equation (7) reflects the observable information that we have on the relationship between the aggregates $y_{i\cdot }$ and the indicators $y_{ij}$ through the observable K-dimensional vector $C_{\cdot j}$. Denoting as ${\hat{y}}_{ij}^{0}$ to the solution in absence of this information, this is given by the indicator $x_{ij}$; i.e., ${\hat{y}}_{ij}^{0}=x_{ij}=\overset{M}{\underset{m=1}{\sum }}q_{ijm}z_{m}$.

Following Golan et al., (1994), the aggregate vectors $y_{i\cdot }$ and $C_{\cdot j}$ are, respectively, row and column margins in a matrix of inter-industry flows. However, the availability of sample (observable) and out-of-sample (unobservable) information could be different in our estimation problem, because in the inter-industry problem it is natural to have known K+T data, but in other estimation problems we only have aggregate information across the dimension of T through $y_{i\cdot }$. For example, if we want to disaggregate the income per capita in each area *i* ($y_{i\cdot }$) into the income per capita of its sub-populations (men and women, population classified by education levels, etc.) being observable the weight of each sub-population on the total population, but not the overall income per capita of each sub-group.

Sometimes the aggregate $C_{\cdot j}$ is not observable and it is replaced by the observation of the weights given to the sub-category *j* in each area *i* ($\theta_{ij}$) that defines the indicator $y_{i\cdot }$ as the weighted sum:(9)$$y_{i\cdot }=\overset{K}{\underset{j=1}{\sum }}y_{ij}\theta_{ij};i=1,\ldots ,T$$

Additionally, the relation between the target indicators $y_{ij}$ and the prior information $x_{ij}$ will be made explicit by means of a functional relationship like:(10)$$y_{ij}=\alpha_{i}+\beta_{ij}x_{ij}+\varepsilon_{ij}$$
and, consequently:(11)$$y_{i\cdot }=\overset{K}{\underset{j=1}{\sum }}\left(\alpha_{i}+\beta_{ij}x_{ij}+\varepsilon_{ij}\right)\theta_{ij};i=1,\ldots ,T$$

Equations (10) and (11) contain the starting point of the traditional approach to spatial disaggregation based on some Distributionally Weighted Regression (DWR) of the type proposed in [13,14]. In Equation (10), the unobservable $y_{ij}$ are defined as a linear function of $x_{ij}$, allowing for slope heterogeneity (note that the $\beta_{ij}$ can be different for each area and sub-class) and an specific area indicator $\alpha_{i}$ plus an error term $\varepsilon_{ij}$. For the estimation of model Equation (10), the same IT-based strategy is followed, by defining for the M possible realizations of each parameter, the support vector $b^{\prime }=\left[b_{1},b_{2},\ldots ,\;b_{M}\right]$ (again common for parameters $\alpha_{i}$ and $\beta_{ij}$) with unknown probabilities $p^{\alpha },p^{\beta }$ to be recovered. The noise components $\varepsilon_{ij}$ are treated in the same ways as in Equation (5).

Once the respective supporting vectors and the a priori probability distributions are set, the DWR estimation can be made in the terms of the following GCE program:(12)$$\begin{array}{c} \underset{p^{\alpha },p^{\beta },W}{Min}D\left(p^{\alpha },p^{\beta },\;W\|q^{\alpha },q^{\beta },\;W^{0}\right)=\\ \overset{M}{\underset{m=1}{\sum }}\overset{T}{\underset{i=1}{\sum }}p_{mi}^{\alpha }ln\left(\frac{p_{mi}^{\alpha }}{q_{mi}^{\alpha }}\right)+\overset{M}{\underset{m=1}{\sum }}\overset{T}{\underset{i=1}{\sum }}\overset{K}{\underset{j=1}{\sum }}p_{mij}^{\beta }ln\left(\frac{p_{mij}^{\beta }}{q_{mij}^{\beta }}\right)+\\ \overset{L}{\underset{l=1}{\sum }}\overset{T}{\underset{i=1}{\sum }}\overset{K}{\underset{j=1}{\sum }}w_{ijl}ln\left(\frac{w_{ijl}}{w_{ijl}^{0}}\right) \end{array}$$
subject to:(13)$$y_{i\cdot }=\overset{K}{\underset{j=1}{\sum }}\left(\overset{M}{\underset{m=1}{\sum }}p_{mi}^{\alpha }b_{m}^{\alpha }+\overset{M}{\underset{m=1}{\sum }}p_{mij}^{\beta }b_{m}^{\beta }x_{ij}+\overset{L}{\underset{l=1}{\sum }}w_{ijl}v_{l}\right)\theta_{ij};\;i=1,\ldots ,T$$
(14)$$\overset{M}{\underset{m=1}{\sum }}p_{mi}^{\alpha }=\overset{M}{\underset{m=1}{\sum }}p_{mij}^{\beta }=\overset{L}{\underset{l=1}{\sum }}w_{ijl}=1;\forall i=1,\ldots ,T;\;j=1,\ldots ,K$$

Both for the parameters and the errors, the supporting vectors usually contain values symmetrically centered on zero. If all the a priori distributions ($q^{\alpha },q^{\beta },\;W^{0}$) are specified as uniform, then the GCE solution reduces to the GME one.

## 3. Unifying the Two Approaches: A Composite Prior Estimator

In this section, we will unify the two previous approaches under a common framework showing that the matrix adjustment problem introduced in [9] is simply a case of a DWR equation (if the available observable information is the same) with not necessarily uniform distributions for $q^{\alpha }$ and $q^{\beta }$. We let out of the discussion the a priori distribution of the errors $W^{0}$ because the uniform solution is the most intuitive. We will base our explanation on the most common case of supporting vectors with M≥2 values distributed symmetrically around zero. 

Note that the GME solution to the DWR problem departs from the specification of a priori distributions that assume that the parameters can take any value as long as they remain in the bounds set in the supports. In contrast, in the solution offered in [9] for the inter-industry flows estimation, no area-specific (row-specific in terms of the problem discussed there) effect was considered and the prior expectation on $y_{ij}$ is given by the corresponding cell $x_{ij}$. These assumptions can be formulated in terms of the a priori distributions used in the DWR approach, which means that both approaches can be treated as particular cases of a general estimation problem. 

The a priori distribution $q^{\alpha }$ can be defined in order to consider the assumption of avoiding any area-specific parameter $\alpha_{i}$ from Equation (10). As opposed to the GME’s solution to the DWR estimation where they are specified as uniform ($q_{}^{\alpha u}$), now we specify an alternative non-uniform distribution ($q_{}^{\alpha n}$) with a point mass at $b_{m}^{\alpha }=0$. Similarly, the a priori distribution $q^{\beta }$ should reflect that the uninformative estimation of $y_{ij}$ is the regressor $x_{ij}$. This non-uniform distribution ($q_{}^{\beta n}$), consequently, should be specified as fulfilling the condition ${\hat{y}}_{ij}^{0}=x_{ij}$, or alternatively:(15)$$\sum_{m=1}^{M}b_{m}^{\beta }q_{ijm}^{\beta n}=1;\;i=1,\ldots ,T;\;j=1,\ldots ,K$$

Appendix A illustrates how specifying such an a priori distribution for the simplest case with *M* = 2 values in the supporting vectors. Having made explicit that, under the same information availability, the two approaches only differ on the a priori distributions specified, it is possible to apply a composite prior estimator that considers both possibilities in the same fashion as in in [10,11]. This estimator is very flexible in the assumptions made on the a priori distributions, given that it allows for including both uniform and non-uniform priors. The estimator it is called Data Weighted Prior (DWP) because it is the information observed which weighs the two alternative priors considered. Furthermore, the authors of [10] prove that its estimates present relatively lower variance than those estimated from a GCE program.

Specifically, the DWP program can be written for our problem as:(16)$$\begin{array}{c} \underset{p^{\alpha },p^{\beta },P^{\gamma },W}{Min}D\left(p^{\alpha },p^{\beta },\;p^{\gamma },W\|q^{\alpha },q^{\beta },\;q^{\gamma },W^{0}\right)=\\ \left(1-\gamma_{i}^{\alpha }\right)\overset{M}{\underset{m=1}{\sum }}\overset{T}{\underset{i=1}{\sum }}p_{mi}^{\alpha u}ln\left(\frac{p_{mi}^{\alpha u}}{q_{mi}^{\alpha u}}\right)+\left(1-\gamma_{ij}^{\beta }\right)\overset{M}{\underset{m=1}{\sum }}\overset{T}{\underset{i=1}{\sum }}\overset{K}{\underset{j=1}{\sum }}p_{mij}^{\beta u}ln\left(\frac{p_{mij}^{\beta u}}{q_{mij}^{\beta u}}\right)+\\ \gamma_{i}^{\alpha }\overset{M}{\underset{m=1}{\sum }}\overset{T}{\underset{i=1}{\sum }}p_{mi}^{\alpha n}ln\left(\frac{p_{mi}^{\alpha n}}{q_{mi}^{\alpha n}}\right)+\gamma_{ij}^{\beta }\overset{M}{\underset{m=1}{\sum }}\overset{T}{\underset{i=1}{\sum }}\overset{K}{\underset{j=1}{\sum }}p_{mij}^{\beta n}ln\left(\frac{p_{mij}^{\beta n}}{q_{mij}^{\beta n}}\right)+\\ \overset{H}{\underset{h=1}{\sum }}\overset{T}{\underset{i=1}{\sum }}p_{hi}^{\gamma \alpha }ln\left(\frac{p_{hi}^{\gamma \alpha }}{q_{hi}^{\gamma \alpha }}\right)+\overset{H}{\underset{h=1}{\sum }}\overset{T}{\underset{i=1}{\sum }}\overset{K}{\underset{j=1}{\sum }}p_{hij}^{\gamma \beta }ln\left(\frac{p_{hij}^{\gamma \beta }}{q_{hij}^{\gamma \beta }}\right)+\\ \overset{L}{\underset{l=1}{\sum }}\overset{T}{\underset{i=1}{\sum }}\overset{K}{\underset{j=1}{\sum }}w_{ijl}ln\left(\frac{w_{ijl}}{w_{ijl}^{0}}\right) \end{array}$$
subject to:(17)$$y_{i\cdot }=\overset{K}{\underset{j=1}{\sum }}\left(\overset{M}{\underset{m=1}{\sum }}p_{mi}^{\alpha }b_{m}^{\alpha }+\overset{M}{\underset{m=1}{\sum }}p_{mij}^{\beta }b_{m}^{\beta }x_{ij}+\overset{L}{\underset{l=1}{\sum }}w_{ijl}v_{l}\right)\theta_{ij};\;i=1,\ldots ,T$$
(18)$$\begin{array}{c} \overset{M}{\underset{m=1}{\sum }}p_{mi}^{\alpha }=\overset{M}{\underset{m=1}{\sum }}p_{mij}^{\beta }=\overset{H}{\underset{h=1}{\sum }}p_{hi}^{\gamma \alpha }=\overset{H}{\underset{h=1}{\sum }}p_{hij}^{\gamma \beta }=\overset{L}{\underset{l=1}{\sum }}w_{ijl}=1;\;\\ i=1,\ldots ,T;\;j=1,\ldots ,K \end{array}$$

The γ parameters are estimated simultaneously with the rest of coefficients of the model. Each γ  measures the weight given to the uniform prior $q^{u}$ for each parameter and it is defined as $\gamma =\sum_{h=1}^{H}b_{h}^{\gamma }p_{h}^{\gamma }$, where $b_{1}^{\gamma }=0$ and $b_{H}^{\gamma }=1$ are, respectively, the lower and upper bound defined in the supporting vectors with H values for these parameters ($b^{\prime }=\left(0,\ldots ,1\right)\to 0\leq \gamma \leq 1$). The a priori probability distributions are always uniform $\left(q_{h}^{\gamma }=\frac{1}{H}\;\right)$ and the same is applied for the errors ($w_{ijl}^{0}=\frac{1}{J}$).

To understand the logic of this estimator, an explanation on the objective function of the previous minimization program is required. Note that Equation (16) is divided in four terms. The last term measures the Kullback divergence between the posterior and the prior probabilities for the noise component of the model. The first term quantifies this divergence between the recovered probabilities and the uniform priors for each coefficient, being this divergence weighted by the corresponding (1−γ). Next, the second element of (16) measures the divergence with the non-uniform priors and it is weighted by γ. The third element in (16) relates to the Kullback divergence of the weighting parameters γ. Equation (16) is minimized subject to the set of constraints present in Equations (16)–(18). Again, the restrictions in (18) ensure that the posterior probability distributions of the estimates and the errors are compatible with the observations, and Equation (18) are just normalization constraints.

## 4. A Numerical Experiment

The numerical simulation compares the performance of the estimation strategies explained previously to estimate a set of latent indicators (T×K). The target will be the unknown elements $\;y_{ij}$ (output per worker, income per capita, etc.) that measure the amount of certain variable $\;z_{ij}$ per unit of other auxiliary variable $\;l_{ij}$. The values of the later are drawn from a normal distribution as $\;l_{ij}\sim N\left(20,2\right)$, which define the weights as $\;\theta_{ij}=l_{ij}/l_{i\cdot }$ We also simulate an observable disaggregated indicator $\;x_{ij}$ drawn as $\;x_{ij}\sim N\left(10,1\right)$ related to our unobservable target $y_{ij}$.

In the context of simulation, we assume that the indicator $\;y_{ij}$ is generated as a convex combination from two possible schemes: (19)$$y_{ij}=\delta \left[\alpha_{i}+\beta_{ij}x_{ij}+\varepsilon_{ij}\right]+\left(1-\delta \right)\left[\eta_{ij}x_{ij}+\varepsilon_{ij}\right];\begin{array}{cc} & i=1,..,T; \end{array}j=1,..,K.$$

This equation contains two sets of slope parameters, namely $\beta_{ij}$ and $\eta_{ij}$, which relate the regressor $x_{ij}$ with the target $y_{ij}$. Furthermore, a fixed area effect $\alpha_{i}$ is also included. These parameters have been arbitrarily set as:(20)$$\begin{array}{c} \alpha_{i}\sim N\left(5,1\right)\\ \beta_{ij}\sim N\left(0,0.1\right)\\ \eta_{ij}\sim N\left(1,0.1\right) \end{array}$$
and they are kept constant along the simulations. The error term $\varepsilon_{ij}$ is drawn as $\varepsilon_{ij}\sim N\left(0,0.1\right)$ and it is generated in each new trial of the experiment.

The first part of the equation ($\alpha_{i}+\beta_{ij}x_{ij}+\varepsilon_{ij}$) shows that $y_{ij}$ can be generated from a process like the one depicted in (16): a linear function of $x_{ij}$ with slope heterogeneity plus a specific area effect (see 11). The second term ($\eta_{ij}x_{ij}+\varepsilon_{ij}$) does not include any specific area indicator and assumes that $y_{ij}$ is exclusively affected by $x_{ij}$ (see 2). Equation (19) includes the scalar δ bounded between 0 and 1 that weighs the two possible sources that generate the variable. If we make δ→1, the first possible mechanism takes over and the contrary happens when we make δ→0. Note that if we set δ=1 we are imposing a data-generating process in line with the assumptions made in the GME program depicted in Equations (12)–(14) for the DWR estimation. On the contrary, if we set δ=0, this is a scenario compatible with the assumptions of non-uniform priors for the parameters that reflected the belief of absence of area-specific effects and a slope parameter close to 1 (labeled as GCE when the simulation results are shown). Any other value of δ between these two extreme cases shows a data-generating process that is not fully incorporated in the priors of either alternative. It is in this type of intermediate situation with the composite prior estimator (labeled as DWP in the simulation results) described in Equations (16)–(18) can be useful, because both priors are considered and we let the data speak for themselves and favor the most realistic one.

The unobservable indicators generated in (20) will be estimated by the three estimation strategies described in the paper (DWR, GCE and DWP estimators) with equal amounts of observable information (the aggregates $y_{i\cdot }=\overset{K}{\underset{j=1}{\sum }}y_{ij}\theta_{ij}$). We have specified a common supporting vector for all the parameters with M=3 points at $b^{\prime }=\left(-10,\;0,\;10\right)$. Similarly, a three-point (H=3) support vector with values 0, 0.5 and 1 has been set for the weighting parameters γ. For the error terms, the support with L=3 values has been chosen, applying the three-sigma rule with uniform a priori weights.

In the experiment, we compare the performance of the three approaches under different scenarios. Three different dimensions (T×K) of the matrix with the target indicators $\;y_{ij}$ have been considered and for each case we set arbitrarily six different values of scalar δ: 0.0; 0.2; 0.4; 0.6; 0.8 and 1.0. In each one of these 18 scenarios, we have carried out 200 trials and computed the mean of the absolute deviation in percentage between our estimates and the real $\;y_{ij}$. Table 1 shows the results: 

Independently of the estimation approach, the numbers on Table 1 show some common patterns to the three of them. The deviations increase with the value of the scalar δ given that high values of this scalar give more weight to the part of the data-generating process that includes an area-specific effect, which makes the $\;y_{ij}$ indicators more difficult to predict. The errors seem more stable regarding the different sizes of the target matrices.

If we pay attention to the comparative performance among the three approaches evaluated in the experiment, the results indicate (not surprisingly) that, for low values of the scalar δ, it seems preferable considering that the GCE approach does not introduce any area-specific effect and considers the regressor $x_{ij}$ as the best prediction in absence of observable information. The longer the value of this scalar, the better the relative performance of the GME-DWR approach (based on a priori uniform distributions). 

The rule of thumb would be, consequently, to use the former when we suspect that no area-specific effect is present (if the second term in Equation (19) dominates) and to favor the latter otherwise (if the first term is more important). In empirical estimation problems, is virtually impossible to know beforehand which one of the two terms is more important. It is in these situations when the use of the composite prior estimator can be helpful. The DWP approach generally outperforms the competing estimators for intermediate values of δ (ranging from 0.4 to 0.8). These medium values indicate some degree of uncertainty about the type of process that generates the data to be estimated. Moreover, the DWP approach can be seen as a conservative solution: even when one of the two parts of the process is clearly dominant (δ=0 or δ=1), the composite prior does not perform much worse than the best of the three options. The losses in terms of prediction, however, can be larger if we choose one single-prior estimator when the other is the best option (see the first and last rows of Table 1).

## 5. An Empirical Application: Obtaining Disaggregated Information on Wages

In order to illustrate the performance of the proposed estimator, it will be applied to solve an empirical problem of disaggregating data of average wages for Spain. The most detailed information about non-agricultural wages in Spain is published in the Wage Structure Survey (*Encuesta de Estructura Salarial*). The complete version of this survey is conducted by the Spanish Statistical Office (*INE*) every four years, being the corresponding to 2010 one of the most recent ones. In intermediate years, however, only partial data are collected and the microdata are not released. If, for example, we want to explore the differences across industries on average wages by gender and type of working day in a year where the complete statistical operation is not conducted, the only information we have are at aggregate level. This situation happens, for example, in 2011, where the only available data on are the aggregates reported in Table 2, which do not allow disaggregated differences between male and female workers to be analyzed depending on the industry they belong to:

In such a context, if the researcher wants to study wage gender gaps across industries it would be necessary to apply an estimation procedure that produces disaggregated values for this specific year, since the official aggregated data do not allow for this type of analysis. The values in Table 2 provide the aggregates required for applying our DWP estimator. Vector y, with dimension (18×1) and elements $y_{i\cdot }$, contains the mean wage for each industry and our estimation target will be the unknown $y_{ij}$ elements, where sub-index j refers to the type of worker (classified into four categories: full-time males, full-time female, part-time male and part-time females). The information in Table 2 is also useful for setting a regressor ($x_{ij}$) for our analysis. In particular, the aggregate mean wages for each type of worker ($x_{i\cdot }$, in the four bottom rows of Table 2) will be used for this purpose, assuming that $x_{i\cdot }=x_{ij},\;j=1,\ldots ,4$. The additional information required to define the weights ($\theta_{ij}$) has been taken from the Spanish Labor Force Survey (EPA) corresponding to that year, where we can find information about the number of workers classified by industry, type of working day and gender. With all this information, the DWP estimator has been applied, specifying identical support vectors as those described in the previous section with the numerical simulation, and the estimates obtained are shown in Table 3:

The aggregate information classified by industry in Table 2 displayed a high variability, ranging from slightly more than EUR 14,000 for the average worker in the Accommodation industry to almost three times higher in Financial and Insurance services. Additionally, the aggregates also showed that the male workers earned more on average than the female workers. Specifically, full-time male workers earned on average around 16% than their female counterparts, whereas this gap was around 11% in the case of part-time workers. This information, however, does not allow for checking if this gender differences on wage keep stable independently on the industry. The estimates obtained by the DWP estimator and reported on Table 3 help to shed some light on this matter. 

According to the outcomes of the estimation, the gender gap for full-time workers is much larger in the case of economic branches related to mining, manufacturing or construction than in service activities. Furthermore, for the specific case of Education and Health and social services activities, we estimate significant positive difference for full-time female workers. Something similar, but to a lesser extent, happens with the case of part-time workers: the mean gender gap in favor of male workers, according to the estimates, is mainly produced by the higher wages received in mining, manufacturing and construction, but in general the activities related to services tend to alleviate this gap. Detecting these differential patterns across industries is possible due to the disaggregated information contained in the estimates, which was partially hidden in the aggregated averages. Additionally, we have explored how robust are the estimates and the patterns found by modifying the supporting vectors, which in turn impact on the priors, as depicted in equation (15). The estimates reported in Table 3 correspond to a case where the support vectors have been defined as $b^{\prime }=\left[-100,0,100\right]$ with M=3 and common for parameters $\alpha_{i}$ and $\beta_{ij}$. Appendix B reports the same estimates as in Table 3, where the support vectors are defined as $b^{\prime }=\left[-10,0,10\right]$ (Table A1) and $b^{\prime }=\left[-1,000,0,1,000\right]$ (Table A2) in order to check if having wider or narrower vectors impacts on the results. Despite some of the minor differences produced by the numerical simulation, the general patterns seem to be robust to this specification. 

## 6. Conclusions

In this paper, we have tackled the problem of providing reliable estimates of a target variable in a set of small geographical areas, by showing that under certain conditions the generalized cross-entropy (GCE) solution for a matrix adjustment problem and the GME estimator of a DWR equation differ only in terms of the a priori information considered. Then, a composite estimator that combines the priors considered in both approaches is proposed and the performance among the three approaches is evaluated throughout Montecarlo experiments. 

The proposed method may represent a new basis to recover estimate at a disaggregate level in presence of: (i) sampling and response errors; (ii) small samples. Within this framework, minimal distributional assumptions are necessary, and a dual loss function is used to take into account both the estimation precision and the prediction objectives. The choice of the prior is data based and endogenously determined and the method provides a simple way of introducing and evaluating prior information in the estimation process. The DWP estimation procedure seem to be a promising alternative model-based estimation technique because the implementation of the method involves minimum outlay of computing, it does not depend on any hypotheses regarding the form of the error distribution in the model, and it produces good results for small-sized samples, especially in the presence of spatial heterogeneity. Finally, theoretical and other non-sample information may be directly imposed on the DWP estimates much more easily than the classic Maximum likelihood and Bayesian estimation techniques. 

The results indicate that for low values of the parameter δ (that measures the weight given to the uniform prior for each parameter), it seems preferable considering the GCE approach that does not introduce any area-specific effect and considers the indicator observed at area level as the best prediction in absence of observable information. The longer the value of this scalar, the better the relative performance of the GME-DWR approach (based on a priori uniform distributions). 

The working of the proposed estimation procedure has been also illustrated by applying the procedure on the estimation of average wages for the Spanish industries in 2011, classified by gender and type of working day. Our results have shown that the DWP estimation has the potential to obtain disaggregated estimates based on minimal assumptions about the data-generating process.

## Figures and Tables

**Table 1 entropy-22-00781-t001:** Results of the numerical experiment (1000 replications): deviation figures.

		Matrix 1 (20×4)	Matrix 2 (50×4)	Matrix 3 (100×4)
γ=0.00	**DWR**	13.126 (0.049) [1.544]	13.642 (0.126) [1.622]	14.837 (0.040) [1.767]
	**GCE**	11.420 (0.126) [1.275]	10.047 (0.054) [1.232]	11.633 (0.038) [1.382]
	**DWP**	11.546 (0.087) [1.321]	11.267 (0.091) [1.352]	12.645 (0.002) [1.494]
γ=0.20	**DWR**	13.697 (0.053) [1.429]	13.667 (0.116) [1.462]	14.791 (0.035) [1.595]
	**GCE**	12.623 (0.131) [1.249]	10.639 (0.044) [1.276]	11.996 (0.044) [1.297]
	**DWP**	12.393 (0.091) [1.248]	11.420 (0.081) [1.233]	12.654 (0.004) [1.356]
γ=0.40	**DWR**	15.307 (0.057) [1.382]	14.357 (0.107) [1.357]	15.381 (0.029) [1.479]
	**GCE**	14.788 (0.136) [1.282]	12.213 (0.035) [1.288]	13.306 (0.049) [1.278]
	**DWP**	14.247 (0.095) [1.243]	12.258 (0.072) [1.175]	13.406 (0.009) [1.282]
γ=0.60	**DWR**	18.565 (0.062) [1.399]	15.922 (0.097) [1.307]	17.152 (0.024) [1.429]
	**GCE**	18.603 (0.141) [1.373]	15.264 (0.025) [1.288]	16.098 (0.055) [1.330]
	**DWP**	17.666 (0.100) [1.300]	14.222 (0.062) [1.182]	15.465 (0.015) [1.278]
γ=0.80	**DWR**	25.047 (0.067) [1.466]	19.109 (0.088) [1.313]	20.898 (0.018) [1.439]
	**GCE**	26.062 (0.145) [1.519]	20.652 (0.016) [1.409]	21.302 (0.060) [1.449]
	**DWP**	24.405 (0.105) [1.409]	18.271 (0.053) [1.255]	19.764 (0.020) [1.341]
γ=1.00	**DWR**	42.350 (0.071) [1.578]	26.659 (0.079) [1.374]	31.769 (0.013) [1.506]
	**GCE**	46.481 (0.149) [1.711]	31.563 (0.006) [1.595]	34.915 (0.066) [1.625]
	**DWP**	42.903 (0.109) [1.564]	27.362 (0.043) [1.385]	31.829 (0.026) [1.466]

Values on each cell report the mean absolute deviation (in %) between the real generated target values and the estimated ones. Values in parentheses show the average bias, on absolute terms (ABIAS), and the figures in brackets show the root of the mean squared errors of the estimates (RMSE).

**Table 2 entropy-22-00781-t002:** Available information on annual wages by industry, type of working day and gender. Wage Structure Survey, 2011.

**Industry**	**Mean Wage (EUR)**
Mining and quarrying industries	29,223
Manufacturing industry	25,308
Supply of electrical energy, gas and steam	50,371
Water supply, sewerage and waste management	25,570
Construction	22,541
Trade and repair of vehicles	19,445
Transport and storage	23,347
Accommodation	14,235
Information and communications	32,491
Financial and insurance activities	41,124
Real estate activities	20,349
Professional, scientific and technical activities	25,350
Administrative and support service activities	16,199
Public administration	27,816
Education	21,565
Health and social services activities	26,058
Arts, recreation and entertainment activities	18,106
Other services	17,035
**Type of Working Day and Gender**	**Mean Wage (EUR)**
Full-time female	23,693
Full-time male	27,596
Part-time female	10,078
Part-time male	11,233

**Table 3 entropy-22-00781-t003:** DWP estimates on disaggregated mean annual wages (EUR) by industry, type of working day and gender, 2011.

Industry	Full-Time, Female	Full-Time, Male	Part-Time, Female	Part-Time, Male
Mining and quarrying industries	13,338	31,307	5311	5840
Manufacturing industry	16,323	29,220	5738	5911
Supply of electrical energy, gas and steam	36,909	55,191	7445	7330
Water supply, sewerage and waste management	14,301	28,675	5419	6135
Construction	12,459	24,134	5239	5987
Trade and repair of vehicles	19,603	23,324	7453	6298
Transport and storage	14,336	26,803	5664	6248
Accommodation	15,473	17,508	6553	6230
Information and communications	23,483	39,877	6741	7326
Financial and insurance activities	38,566	46,664	7946	6078
Real estate activities	21,301	23,487	7301	6299
Professional, scientific and technical activities	25,022	29,926	7984	6565
Administrative and support service activities	17,383	20,534	9142	6290
Public administration	24,433	32,196	6269	6117
Education	25,838	21,708	8396	6720
Health and social services activities	31,832	20,406	9049	6078
Arts, recreation and entertainment activities	17,460	24,232	8094	8778
Other services	19,600	18,896	7537	6116

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
