# Peer review of "Entropy-Based Solutions for Ecological Inference Problems: A Composite Estimator"

_entropy, 2020, doi:10.3390/e22070781_

Round 1
Reviewer 1 Report
The paper "Entropy-based solution for Ecological Inference problems: a composite estimator" is quite interesting and innovative, according to my point of view. The literature on GCE and GME does not show great progress in the last years, given the difficulty to implement those estimation methods. The authors present a very interesting and useful method for distributional weighted regression.
I believe that this paper would increase its significance of content if the contributions are better explained. With the same objective, the empirical application to non-agricultural wages on Spain, need more interpretation and intuition.
The main problems about GCE and GME remain, and authors should clarify both:
-How to choose priors p and v? How can priors influence the estimation results?
-Robustness of the results given different priors?
-The authors should make available the code used and clarify the base software.
Author Response
The paper "Entropy-based solution for Ecological Inference problems: a composite estimator" is quite interesting and innovative, according to my point of view. The literature on GCE and GME does not show great progress in the last years, given the difficulty to implement those estimation methods. The authors present a very interesting and useful method for distributional weighted regression.
I believe that this paper would increase its significance of content if the contributions are better explained.
- With the same objective, the empirical application to non-agricultural wages on Spain, need more interpretation and intuition.
Reply: thanks for the suggestion. The revised version includes additional comments on this survey and how the proposed estimator can be helpful to recover information not directly accessible to the researcher.
The main problems about GCE and GME remain, and authors should clarify both:
- How to choose priors p and v? How can priors influence the estimation results?
- Robustness of the results given different priors?
Reply: we took the liberty of giving a common reply to these two comments, since they are strongly connected. The reviewer is right on his/her remark on the potential impact of the choice of priors in the results. This has been an issue for this class of GCE estimators in situations where this choice is not clear and depends on the “taste” of the researcher. However, the estimator presented here does not need to specify subjective prior, since they depend on the specification of the support vectors, as explained on page 6 (lines 142 to 155 approximately) and illustrated in Appendix A1.
This, of course, does not totally avoid the problem, since the choice of the elements in these supporting vectors is immediately translated to the priors, which can impact on the estimates. Following the suggestion by the reviewer, in this revised version we include additional results that show the estimates of the empirical application under different supports for the beta parameter. We do not include variability in the support for the error terms because we are applying the traditional 3-sigma rule. The results reported in the new version of Table 3 show that, are expected, the results are not completely unsensitive to the specifications of these supports, but the general conclusions seem robust.
- The authors should make available the code used and clarify the base software.
Reply: We include as supplemental material in this revised version the GAMS code used.
We hope that you find this revised version satisfactory. Thanks again for your constructive comments.
GAMS CODE:
*Define the dimensions and read the data =======================================
sets
ind industry /s1*s18/
class typy of contract and gender /c1*c4/
alias (class,j)
alias (ind,i) ;
$CALL GDXXRW.EXE EES.xls par=data rng=matrix!b4:g23 Rdim=1 Cdim=1
Parameter data(*,*);
$GDXIN EES.gdx
$LOAD data
$GDXIN
Parameter R(i) row sums
/
s1 1204007.0
s2 58330131.9
s3 3969267.1
s4 3336918.9
s5 31397358.9
s6 56939680.7
s7 20612704.3
s8 19813056.2
s9 16524942.9
s10 18526461.1
s11 1941304.1
s12 20822835.0
s13 14592185.3
s14 39298572.0
s15 25387980.0
s16 37177348.1
s17 5638295.6
s18 6597721.3
/ ;
Parameter C(j) column sums
/
c1 23692.76
c2 27595.54
c3 10077.81
c4 11232.73
/;
Table L(i,j)
c1 c2 c3 c4
s1 3.2 36.9 0.6 0.5
s2 495.2 1697.4 78.3 33.9
s3 14.5 61.9 1.4 1
s4 17.7 106.2 2.8 3.8
s5 77 1244.2 25.5 46.2
s6 1114.1 1366.9 347.7 99.6
s7 132.1 683.4 33.5 33.9
s8 532.5 545.1 222.8 91.5
s9 144 322.1 21.2 21.3
s10 191.2 235 20.9 3.4
s11 40.8 41.8 9.8 3
s12 325.9 398.6 73.1 23.8
s13 292.5 355.5 216.6 36.2
s14 543.5 793.5 53.6 22.2
s15 624 353.3 150.2 49.8
s16 916.5 303.9 184.5 21.8
s17 78.6 145.7 42.8 44.3
s18 189.1 124.6 61.1 12.5 ;
Parameter mean_wage(j)
/
c1 23693
c2 27596
c3 10078
c4 11233
/;
Parameter X(i,j);
loop(i,X(i,j)=mean_wage(j));
Parameter sum_L(i);
sum_L(i)=sum(j,L(i,j));
Parameter weight(i,j);
weight(i,j)=L(i,j)/sum_L(i) ;
*====== these are the supports ================================================
sets m 'theta support' /b1*b3/
k 'error support' /v1*v3/
h gamma support /h1*h3/
Scalar EME; EME=10;
Parameter support(m);
support("b1")=-EME;
support("b2")=0;
support("b3")=EME;
parameter rule(k) '3 sigma rule'
/v1 -3
v2 0
v3 3/
scalar T; T=card(i);
scalar mean; mean=sum(i,R(i))/T;
scalar std; std=sqrt(sum(i,sqr(R(i)-mean))/(T-1));
Parameter v(k) 'error support' ;
v(k)=rule(k)*Std ;
parameter g(h) 'support for gamma'
/h1 0
h2 0.5
h3 1/ ;
*====== these are the uniform priors ==========================================
Parameter qo_U(m,i);
loop((m,i),qo_U(m,i)=(1/3));
Parameter q1_U(m,i,j);
loop((m,i,j),q1_U(m,i,j)=(1/3));
*====== these are the non-uniform priors ======================================
Parameter qo_n(m,i);
loop(i,
qo_n("b2",i)=0.999;
qo_n("b1",i)=(1-qo_n("b2",i))/2;
qo_n("b3",i)=1-qo_n("b1",i)-qo_n("b2",i);
);
Parameter q1_n(m,i,j);
loop((i,j),
q1_n("b1",i,j)=(EME-1)/(3*EME);
q1_n("b2",i,j)=(EME-1)/(3*EME);
q1_n("b3",i,j)=(EME+2)/(3*EME);
);
*===============================================================================
*===============================================================================
*===============================================================================
*The optimization starts here ==================================================
Variables
p0_DWP(m,i)
p1_DWP(m,i,j)
w_DWP(k,i,j)
p_g_alfa(h,i)
p_g_beta(h,i,j)
local_DWP(i,j)
gamma_alfa(i)
gamma_beta(i,j)
WKL;
Equations
Unityw_DWP(i,j)
Unityp0_DWP(i)
Unityp1_DWP(i,j)
Unityp_g_alfa(i)
Unityp_g_beta(i,j)
def_local_DWP(i,j)
res1_DWP(i)
def_gamma_alfa(i)
def_gamma_beta(i,j)
objec;
*=== these are the equations defining the parameters ===========================
def_local_DWP(i,j).. (local_DWP(i,j))=e=sum(m,p0_DWP(m,i)*support(m))+
sum(m,p1_DWP(m,i,j)*support(m))*(X(i,j))+
sum(k,w_DWP(k,i,j)*v(k));
def_gamma_alfa(i).. gamma_alfa(i)=e=sum(h,p_g_alfa(h,i)*g(h));
def_gamma_beta(i,j).. gamma_beta(i,j)=e=sum(h,p_g_beta(h,i,j)*g(h));
*====== this is the objective function==========================================
Objec.. WKL =e=sum(i,gamma_alfa(i)*(sum((m),p0_DWP(m,i)*log(p0_DWP(m,i)/qo_n(m,i)))))+
sum((i,j),gamma_beta(i,j)*(sum((m),p1_DWP(m,i,j)*log(p1_DWP(m,i,j)/q1_n(m,i,j)))))+
sum(i,(1-gamma_alfa(i))*(sum((m),p0_DWP(m,i)*log(p0_DWP(m,i)/qo_u(m,i)))))+
sum((i,j),(1-gamma_beta(i,j))*(sum((m),p1_DWP(m,i,j)*log(p1_DWP(m,i,j)/q1_u(m,i,j)))))+
sum((h,i),p_g_alfa(h,i)*log(p_g_alfa(h,i)/(1/3)))+
sum((h,i,j),p_g_beta(h,i,j)*log(p_g_beta(h,i,j)/(1/3)))+
sum((k,i,j),w_DWP(k,i,j)*log(w_DWP(k,i,j)/(1/3)));
*====== these are the constraints =============================================
Unityw_DWP(i,j).. sum(k,w_DWP(k,i,j))=e=1;
Unityp0_DWP(i).. sum(m,p0_DWP(m,i))=e=1;
Unityp1_DWP(i,j).. sum(m,p1_DWP(m,i,j))=e=1;
Unityp_g_beta(i,j).. sum(h,p_g_beta(h,i,j))=e=1;
Unityp_g_alfa(i).. sum(h,p_g_alfa(h,i))=e=1;
res1_DWP(i).. (R(i)/sum_L(i))=e=sum(j,(local_DWP(i,j))*weight(i,j));
*==== set the intial values and bounds =========================================
local_DWP.l(i,j)=x(i,j) ;
p0_DWP.lo(m,i)=0.000001 ;
p1_DWP.lo(m,i,j)=0.0000001 ;
w_DWP.lo(k,i,j)=0.0000001 ;
p_g_alfa.lo(h,i)=0.0000001 ;
p_g_beta.lo(h,i,j)=0.0000001 ;
*====== the program is compiled and solved here ===============================
model DWP /Objec,unityp0_DWP,unityp1_DWP, unityw_DWP, def_local_DWP, res1_DWP,
Unityp_g_beta, Unityp_g_alfa, def_gamma_alfa, def_gamma_beta /;
DWP.scaleopt=1 ;
solve DWP using nlp minimizing WKL;
OPTION ITERLIM =90000;
OPTION RESLIM =900000;
*===============================================================================
*===============================================================================
*===============================================================================
*====== the solutions are reported here =======================================
parameter sol_DWP(i,j); sol_DWP(i,j)=(local_DWP.l(i,j));
display sol_DWP;

Reviewer 2 Report
Please see the attached report.

Author Response
Dear reviewer,
Thanks so much for your positive opinions on the manuscript. We have submitted a revised version with some clarifications, as requested by Reviewer #1 that includes:
- the GAMS code to run the empirical application.
- a more detailed explanation on the choice of the priors.
- additional information about the wage survey used in the empirical application.
We hope that you are positive as well with this revised version.
Our best regards,
The authors.